# Lifetime Weight Course as a Phenotypic Marker of Severity and Therapeutic Response in Patients with Eating Disorders

**DOI:** 10.3390/nu13062034

**Published:** 2021-06-13

**Authors:** Zaida Agüera, Cristina Vintró-Alcaraz, Isabel Baenas, Roser Granero, Isabel Sánchez, Jéssica Sánchez-González, José M. Menchón, Susana Jiménez-Murcia, Janet Treasure, Fernando Fernández-Aranda

**Affiliations:** 1Centro de Investigación Biomédica en Red Fisiopatología Obesidad y Nutrición (CIBERobn), Instituto de Salud Carlos III, 08907 L’Hospitalet de Llobregat, Spain; cvintro@bellvitgehospital.cat (C.V.-A.); ibaenas@bellvitgehospital.cat (I.B.); Roser.Granero@uab.cat (R.G.); isasanchez@bellvitgehospital.cat (I.S.); sjimenez@bellvitgehospital.cat (S.J.-M.); 2Department of Psychiatry, University Hospital of Bellvitge, 08907 L’Hospitalet de Llobregat, Spain; jsanchezg@bellvitgehospital.cat (J.S.-G.); jmenchon@bellvitgehospital.cat (J.M.M.); 3Psychiatry and Mental Health Group, Neuroscience Program, Institut d’Investigació Biomèdica de Bellvitge—IDIBELL, 08907 L’Hospitalet de Llobregat, Spain; 4Department of Public Health, Mental Health and Maternal-Child Nursing, School of Nursing, University of Barcelona, 08907 L’Hospitalet de Llobregat, Spain; 5Departament de Psicobiologia i Metodologia de les Ciències de la Salut, Universitat Autònoma de Barcelona, 08193 Barcelona, Spain; 6Department of Clinical Sciences, School of Medicine and Health Sciences, University of Barcelona, 08907 L’Hospitalet de Llobregat, Spain; 7Centro de Investigación Biomédica en Red de Salud Mental (CIBERSAM), Instituto de Salud Carlos III, 08907 L’Hospitalet de Llobregat, Spain; 8Department of Psychological Medicine, Institute of Psychiatry, Psychology and Neuroscience, King’s College London, London WC2R 2LS, UK; janet.treasure@kcl.ac.uk

**Keywords:** body mass index (BMI) profiles, eating disorders, obesity, treatment outcome

## Abstract

The association between lifetime weight fluctuations and clinical characteristics has been widely studied in populations with eating disorders (ED). However, there is a lack of literature examining the potential role of weight course as a transdiagnostic factor in ED so far. Therefore, the aim of this study is to compare ED severity and treatment outcomes among four specific BMI profiles based on BMI-trajectories across the lifespan: (a) persistent obesity (OB-OB; (*n* = 74)), (b) obesity in the past but currently in a normal weight range (OB-NW; *n* = 156), (c) normal weight throughout the lifespan (NW-NW; *n* = 756), and (d) current obesity but previously at normal weight (NW-OB; *n* = 314). Lifetime obesity is associated with greater general psychopathology and personality traits such as low persistence and self-directedness, and high reward dependence. Additionally, greater extreme weight changes (NW-OB and OB-NW) were associated with higher psychopathology but not with greater ED severity. Higher dropout rates were found in the OB-OB group. These results shed new light on the BMI trajectory as a transdiagnostic feature playing a pivotal role in the severity and treatment outcome in patients with ED.

## 1. Introduction

Eating disorders (ED) and obesity have frequently been considered as part of the same continuum of so-called extreme weight conditions [1,2,3]. This continuum is reinforced because both pathologies share risk and maintenance factors that have been widely described in the literature [4,5,6]. Furthermore, genetic factors underlying body mass index (BMI) have been associated with disordered eating behaviors and related cognitions, and these associations have also been mediated by BMI [7]. Among the different ED listed in the DSM-5 [8], binge eating disorder (BED) is the one with the highest prevalence of comorbid obesity [9,10] followed by bulimia nervosa (BN) [11]. Villarejo et al. [11] found that almost 30% of female patients with ED had lifetime obesity, and those patients were characterized by later age of onset, longer duration of the disorder, higher minimum and maximum-ever BMI, and higher eating-related and general psychopathological severity. Similarly, a continuum of severity has been described in which patients with obesity and BN show the highest symptomatology and psychopathology, followed by BED with obesity, with obesity without ED being the least severe [12].

The personality profiles of individuals with overweight or obesity have been widely reported in the literature, both in patients with [13,14] and without ED [15]. Patients with ED and lifetime obesity often present personality profiles characterized by a higher harm avoidance and lower scores in persistence, self-directedness, and cooperativeness than ED patients without obesity [11,12]. In addition, a systematic review on personality traits in obesity identified that high scores on reward sensitivity, impulsivity, and neuroticism may act as risk factors, whereas high self-directedness, persistence, and self-control would act as protective factors [15].

Weight trajectories [16] and fluctuations [17,18] have also been associated with disordered eating behaviors and may be of relevance as risk and maintaining factors [17,18]. Frequent weight fluctuations suggest some degree of dysregulation of weight homeostasis [19]. The difference between the premorbid weight before the onset of ED and current weight has been described as a risk factor for bulimic psychopathology [18,20]. It also appears as a predictor of weight gain during therapy [20,21] and poorer treatment outcomes [22]. Striegel-Moore et al. [23] found a rapid increase in weight trajectory two years prior to the onset of BED. Ivezaj et al. [24] also found that patients with BED and obesity reported a significant weight gain during the year before seeking treatment and this was associated with higher relapse rates, greater ED and affective psychopathology [24,25,26]. Furthermore, a large body of research has revealed that some ED-related characteristics such as emotional eating, binge eating behaviors, poor body image, and high body dissatisfaction are associated with weight fluctuations in patients with ED [27,28]. Indeed, some authors have described above-average weights and more fluctuation in adolescents prior to the onset of the ED (i.e.,) [29]. Likewise, severe weight cycling was more prevalent among adult women with obesity and was associated with higher reward sensitivity, and depressive-related symptomatology, and a higher prevalence of BED [30].

The relationship between weight suppression (WS) (defined as the difference between the highest adult weight and the current weight) and ED has also been the subject of interest in the literature. However, it is difficult to draw firm conclusions as the evidence is mixed. Some studies found no associations between WS and clinical variables [31,32], whereas others found that WS was related to more severe ED symptomatology, greater depression, poorer prognosis, and greater weight gain at post treatment [17,33,34,35].

To our knowledge, no study has examined groups of patients with ED based on lifetime weight trajectories. Therefore, the main goal of the present study was to examine whether obesity across the lifespan might be a transdiagnostic marker of ED severity and treatment outcome. A clinical sample of people with ED was post hoc distributed into four BMI profiles according to the period of obesity over adulthood: (a) with lifespan obesity (OB-OB), (b) with past obesity but currently normal weight (OB-NW), (c) with normal weight throughout their lifespan (NW-NW), and (d) with previous normal weight but current obesity (NW-OB). Therefore, two substudies were conducted. The first cross-sectional substudy aimed (1) to examine whether the different diagnostic categories of EDs are differentially distributed across the BMI profiles and (2) to compare the BMI profiles in terms of motivational stage, ED severity, general psychopathology, personality traits, and impulsive behaviors. The aim of the second prospective substudy was to examine whether the BMI profile predicted treatment outcome.

We hypothesized that the prevalence of patients with BED would be greater in BMI profiles with obesity. A second hypothesis was that patients with increased lifetime weight changes (i.e., OB-NW and NW-OB) would exhibit greater ED symptomatological and psychopathological severity, as a worse treatment outcome.

## 2. Materials and Methods

### 2.1. Participants

The clinical sample consisted of 1300 adult patients with ED. All the patients were consecutive referrals for assessment and treatment to the EDs Unit, Department of Psychiatry at the Bellvitge University Hospital (Barcelona, Spain). The sample was composed of 82 males and 1218 females meeting the criteria for BN (*n* = 719), BED (*n* = 211), and OSFED (*n* = 370). Patients admitted before 2013 were originally diagnosed according to DSM-IV-TR criteria [36]. All diagnoses were recoded post hoc using DSM-5 criteria [8].

The longitudinal substudy was conducted with 500 of the patients from the first substudy (91.6% females; 8.4% males) with available post-treatment data. Although differences were observed between the participants included and those not included in the longitudinal substudy in terms of diagnosis, sex, educational level, and employment status, it should be noted that they did not show significant differences in terms of the main clinical variables such as group distribution, age, age of onset of the ED, and symptomatological ED severity (see Appendix A).

The following exclusion criteria were applied to both substudies: (a) age below 18 years old; (b) having a diagnosis of AN or presenting with a BMI below 18.5 kg/m^2^; (c) currently overweight (BMI: 25–29.9 kg/m^2^). The last two exclusion criteria were made according to a clinical consensus to suit the standardized definitions of normal weight and obesity and to cluster the empirical groups accordingly. Appendix A includes the flowchart with the sampling procedure. Additional analyses confirmed that there was no methodological bias since there were no significant differences between included and nonincluded participants in the distribution of the main variables such as sociodemographic variables (sex: *p* = 0.065, education level: *p* = 0.127, marital status: *p* = 0.288, and age: *p* = 0.074), age of onset of the disorder (*p* = 0.986), and psychopathological severity (SCL-90R PST: *p* = 0.074, SCL-90R GSI: *p* = 0.075, and SCL-90R PSDI: *p* = 0.074).

### 2.2. Assessment

Sociodemographic and clinical data were obtained by means of a face-to-face semistructured interview based on the SCID-5 [37] administered by clinical psychologists and psychiatrists specialized in ED. During this clinical interview, data on the presence of certain impulsive behaviors, such as nonsuicidal self-injury (NSSI) behaviors, suicidal ideation and/or attempts, alcohol abuse, and drug abuse were also retrieved from specific questions that have been previously used in previous research [38,39]. The evolution of weight was recorded by asking about the minimum and maximum weight attained throughout adulthood and at what age they reached this weight, as well as the current weight at the time of assessment. Additionally, the following commonly applied questionnaires in the field of ED were administered:

The Eating Disorder Inventory-2 (EDI-2) [40] is a 91-item self-reported questionnaire that assesses 11 ED-related cognitive and behavioral domains. A total score is also provided to report overall ED severity. This instrument has been validated in a Spanish population [41]. In the current sample, the internal consistency was excellent (α = 0.948).

The Symptom Checklist-90-Revised (SCL-90-R) [42] contains 90 items that measure 9 primary psychopathological dimensions: somatization, obsession–compulsion, interpersonal sensitivity, depression, anxiety, hostility, phobic anxiety, paranoid ideation, and psychoticism; and includes three global indices: global severity index (overall psychological distress), positive symptom distress index (the intensity of symptoms), and a positive symptom total (self-reported symptoms). This scale has been validated in a Spanish population [43]. In the present study, internal consistency was excellent (α = 0.976).

The Temperament and Character Inventory-Revised (TCI-R) [44] is a 240-item self-reported questionnaire that measures seven dimensions of personality: four temperament dimensions (harm avoidance, novelty seeking, reward dependence, and persistence) and three character dimensions (self-directedness, cooperativeness, and self-transcendence). The Spanish validation was carried out by Gutierrez-Zotes et al. [45]. Our internal consistency ranged from α = 0.797 to α = 0.893.

The motivation stage of change was evaluated by means of a visual analog scale, ranging from 0 to 8, which assessed the following five aspects: (1) subjective desire to receive treatment, (2) need for treatment, (3) perceived impairment, (4) self-concern, and (5) parental concern. Higher scores indicated greater worry and motivation to change. This scale has been previously described and applied in other studies [46].

### 2.3. Treatment

Treatment consisted of 16 weekly group outpatient sessions of cognitive behavioral therapy (CBT). There was a total of 8–10 patients per group. Although patients with BN, BED, and OSFED were placed in separate groups of therapy, all the treatment groups were based on the same CBT program. This program and its complementary material have already been manualized and published in Spanish [47] with demonstrated effectiveness [48,49,50,51].

Patients were reevaluated at discharge and categorized into the following DSM-5 categories [8]: full remission (total absence of symptoms meeting diagnostic criteria for at least 4 consecutive weeks), partial remission (substantial symptomatic improvement but with residual symptoms), and nonremission (poor outcome or exacerbation of symptoms). These categories were used to assess treatment outcomes in previously published studies [48,50,52,53,54]. Voluntary treatment discontinuation was categorized as dropout (i.e., not attending treatment for at least three consecutive sessions).

In accordance with the Declaration of Helsinki, the present study was approved by the Ethics Committee of our institution (The Clinical Research Ethics Committee (CEIC) of the Bellvitge University Hospital). All participants provided signed informed consent.

### 2.4. Statistical Analyses

Statistical analysis was carried out with Stata16 (StataCorp, College Station, TX, USA) LLC for windows [55]. The comparison between the four groups of the study (OB-OB, OB-NW, NW-NW, and NW-OB) was based on chi-square tests (χ^2^) for categorical variables and analysis of variance (ANOVA) for quantitative variables.

The effect size for the difference between means was estimated through the standardized Cohen’s *d* coefficient, considering null effect size for *|d|* < 0.20, low-poor for *|d|* > 0.20, moderate-medium for *|d|* > 0.50, and large-high effect for *|d|* > 0.80) [56]. The effect size for the difference between proportions was estimated through the standardized Cohen’s *h* coefficient, which is interpreted similarly to Cohen’s *d* measure and calculated through the arcsine transformation of the rates registered in each group (null effect size is considered for *|h|* < 0.20, low-poor for *|h|* > 0.20, moderate-medium for *|h|* > 0.50, and large-high for *|h|* > 0.80) [57].

An increase in Type-I error due to multiple significance tests was controlled with the Finner method [58], a family-wise error rate (FWER) stepwise procedure, which has proved more powerful than the classical Bonferroni correction. When controlling the *k*-FWER, a fixed number of k-1 of erroneous rejections is tolerated, and under the assumption that all the null hypotheses are equal, controlling the FWER at level α is equivalent to the problem of combining the original-unadjusted *p*-values to obtain single testing for the null hypothesis (H_0_), which is at level α. For example, from a procedure R that controls the FWER at level α is equivalent to derive a single testing procedure of level α by rejecting the H_0_ whenever R(*p*) is not empty (that is, whenever R(p) rejects at least one hypothesis). In practice, the Finner method is employed by adjusting the rejection criteria for each of the individual hypotheses fixing the FWER no higher than a certain prespecified significance level. The procedure starts sorting the *p*(unadjusted)-values (p1, …, pk), achieved in k-independent null-hypothesis tests, into the order of lowest to highest. Then, the next algorithm is used: p(adjusted) = (1 − (1-p(unadjusted))^(total tests/position within the ordered tests).

## 3. Results

### 3.1. Comparison between the Groups for Sociodemographics, BMI, and Motivational Measures

Figure 1 displays the diagnostic profile within each grouping. The first block of Table 1 includes cluster, sociodemographic and diagnostic information type. The OB-OB BMI profile included mainly patients with BED, whereas patients with BN and OSFED diagnoses were within the OB-NW and NW-NW groups. The fourth BMI profile (i.e., NW-OB) consisted mainly of the BN and BED diagnostic types.

The group with the highest prevalence of men was OB-NW, followed by OB-OB and NW-OB, only a small percentage of NW-NW patients were men.

Higher levels of educational achievement were attained in the NW-NW and NW-OB groups. A higher proportion of the NW-OB patients were married. Those in the OB-OB group were more often employed, and those in the NW-NW more often studying.

The NW-OB and OB-OB groups considered themselves to be most symptomatic and in need of treatment. Family members of NW-NW and OB-NW groups expressed most concern.

### 3.2. Comparison between the Groups for Clinical Measures

Table 2 contains the comparison between the groups for the clinical profiles. People in the NW-NW group were younger, with an earlier age of onset and shorter duration of the disorder. Those in the NW-OB group were older, with a later age of onset and longer duration.

The highest frequency of binges was noted within the NW-OB group, while the highest frequency of vomiting was noted in the OB-NW group. The OB-OB group had the lowest number of binges and vomiting episodes.

The highest eating psychopathology (EDI-2 total score) was in the OB-OB group, followed by NW-OB, OB-NW, and NW-NW groups. The most severe general psychopathological state (SCL-90R scores) was related to NW-OB, followed by OB-NW.

The OB-OB group had higher levels of harm avoidance, reward dependence, cooperativeness, and self-transcendence. The NW-NW group had higher levels of novelty seeking, persistence, and cooperativeness. The OB-NW group had lower levels of novelty seeking, reward dependence, and cooperativeness. The NW-OB group had higher levels of harm avoidance.

### 3.3. Comparison between the Groups for Impulsive Behaviors

NSSI behavior, suicidal ideation and attempts, and substance use/abuse are shown in Table 3. The highest proportions of patients who reported NSSI behavior were in the OB-NW group, followed by NW-NW, NW-OB, and OB-OB groups. The highest prevalence of suicidal behavior (ideation and attempts) was registered among NW-OB and OB-NW groups. The lowest prevalence of alcohol and drug consumption was registered among the OB-OB group, while OB-NW and NW-NW groups reported the highest prevalence.

### 3.4. Association between Age of Onset of ED and Age of Maximum BMI

The age of the maximum BMI and age of onset of the ED are shown in Table 4, and a scatterplot of these data is shown in Figure 2. Most patients in the BMI profile of NW-OB and OB-OB registered the maximum BMI after the onset of the ED. Among OB-NW patients, 35.2% registered maximum-ever BMI prior to the onset of ED, 27.2% coinciding with the onset of ED, and 37.6% after the onset of ED. Among NW-NW patients, 24.6% registered maximum-ever BMI prior to the onset of the ED, 22.9% coinciding with the onset of ED, and 52.5% after the onset of ED.

### 3.5. Comparison of the CBT Outcomes between the BMI Profiles

Table 5 shows the distribution of the CBT outcomes between the groups (see also Figure 3). The OB-OB BMI profile registered the highest prevalence of dropouts and the lowest of non-remission. The NW-NW BMI profile presented the highest prevalence of non-remission. OB-NW and NW-OB groups achieved the highest prevalence of partial- or full-remission.

## 4. Discussion

The present study sought to address an important gap in the literature by examining whether the weight history of patients with ED could be a transdiagnostic marker of severity and treatment outcome. We analyzed the psychopathology and dysfunctional personality profiles of the different groups. We also examined whether these BMI profiles had a different response to therapy.

As expected, the main finding was that most patients with BED were in the BMI profile with current obesity. These results are in line with previous studies suggesting that BED is strongly associated with excessive body weight gain [11,59], due to the high-calorie overconsumption in absence of compensatory behaviors and the sedentary lifestyle frequently reported by these patients [10,60]. While most patients with OSFED were mainly represented in the BMI profiles with current normal weight (OB-NW and NW-NW), the patients with BN were more heterogeneous, and they were cited in all the BMI profiles but were mainly normal weight.

Patients with current obesity (i.e., those in OB-OB and NW-OB) reported the highest levels of motivation for change, specifically greater concern, and subjective intensity of their ED, and a higher desire for treatment. It may suggest that they consider their obesity related to the ED and, therefore, they are more motivated to seek treatment. This hypothesis would be reinforced by our results showing that most patients in the OB-OB and NW-OB BMI profiles recorded the maximum-ever BMI after the onset of the ED. On the other hand, it should be noted that these patients also had a longer duration or chronicity of the disorder, which has previously been related to increased motivation and perceived need for treatment [61].

The second main objective was to examine the clinical differences between the groups based on BMI changes over the lifetime. Our findings are partially in agreement with previous research on weight fluctuations. Consistent with previous studies, we found an association between lifetime weight changes (i.e., NW-OB and OB-NW) and a more severe general psychopathological state [24]. However, in contrast to other studies [24,25], we found no association between weight fluctuations and greater ED-related symptomatology. Nevertheless, according to previous research [11], our results corroborate that, overall, patients with lifetime obesity report greater ED and general psychopathology, compared to those without a history of obesity (namely NW-NW). This might suggest that lifetime obesity, rather than weight fluctuations, is associated with greater psychopathology, regardless of whether the obesity was before or after the development of the ED. Our findings show that patients who have never had obesity had higher novelty seeking, and were also higher in persistence and self-directedness than those with lifetime obesity. These findings are similar to those reported by Villarejo et al. [11] in which a more dysfunctional personality profile (characterized by high harm avoidance, and low scores on persistence, self-directedness, and cooperativeness) was described in patients with ED and lifetime obesity. Similarly, high persistence, self-directedness, and self-control have been identified as protective factors for weight gain or obesity development [15]. In addition, this research went a step further and identified the lowest scores on reward dependence and novelty seeking in the group of patients with previous obesity but current normal weight (i.e., OB-NW). Low scores on reward dependence are related to being independent, not influenced by others, nonconformist, socially detached, and insensitive to social pressures. Therefore, although this finding may seem striking, it is in line with a previous study suggesting that people who do not require social support and are more self-confident are more likely to achieve self-directed weight loss [62]. This finding supports the use of therapeutic tools targeted at improving self-reliance, especially in patients with lifetime obesity. On the other hand, our results reveal that patients who achieved a normal weight coming from obesity were those with the lowest scores on novelty seeking. This is consistent with a prior study suggesting that low novelty seeking was associated with weight loss in patients seeking treatment for obesity [63]. Therefore, this finding suggests that using techniques to reduce impulsivity would be useful in the treatment of patients with ED and obesity [64].

Our results also reveal that patients with more extreme weight changes across the lifespan (i.e., OB-NW and NW-OB) endorsed a higher frequency of suicidal ideation and attempts. These results are in line with previous studies reporting more depressive-related symptomatology and severe psychopathology in patients with lifetime weight fluctuations [22,24,30]. The OB-NW profile also engaged in more NSSI behaviors. Additionally, this group had the highest frequency of vomiting episodes, which is in line with previous findings suggesting that NSSI is strongly related to purging behaviors and both may serve similar functions in terms of emotion regulation [65]. The lowest prevalence of substance consumption (alcohol and drugs) was registered among OB-OB. As this BMI profile had the lowest frequency of vomiting, our results are also in agreement with previous studies that found a relationship between a higher frequency of purging behaviors and higher substance use [66,67].

Finally, the longitudinal data indicate that patients in both OB-NW and NW-OB groups had the best treatment outcomes, which is inconsistent with the previous literature that had found a relationship between greater weight fluctuations and lower therapeutic adherence and worse treatment outcome [17]. Patients in the OB-OB BMI profile had the highest dropout rates. This novel and noteworthy result calls into question the findings of previous studies reporting that patients with BED had rapid symptoms remission but also high dropout rates, compared to BN [48]. This previous study suggested that patients with BED, most of them with obesity, dropped out more frequently because their desire to lose weight was not addressed by standard CBT. In addition, the current research expands these findings and indicates differences within this type of patient. Although patients with BED were represented in both NW-OB and OB-OB BMI profiles, only those in the latter group presented a higher prevalence of dropout. One possible rationale could be that patients who have developed obesity after the ED onset may consider their weight gain as a consequence of the disorder and dependent on their recovery. Therefore, they may exhibit greater therapeutic adherence.

### Limitations and Strengths

The present study should be considered within the context of several limitations. First, retrospective and self-report data collection (mainly regarding maximum and minimum weight) may limit the validity and reliability of our results. Participants who may have been underweight (BMI < 18.5 kg/m^2^) in the past were not excluded in this study because of the difficulty of interpreting retrospective reports of age-associated BMI changes and the unavailability of height data. Further studies should exclude participants with a lifetime BMI less than 18.5 by controlling for weight and height at each time point. In addition, although we asked for the age of onset of ED and age of maximum and minimum BMI, our results do not allow us to state that weight changes are a cause or a consequence of the disorder. Second, the low representation of males did not allow for meaningful sex-related comparisons. However, this was representative of the proportion we routinely observe in clinical practice. Third, the motivational scale has not been validated, although it has been used in previous studies [46,49,61]. In this line, further studies should include validated instruments to measure the motivation stage of change. Finally, findings from the longitudinal substudy were based on symptomatological remission after the therapy but not recovery (which requires a long period of abstinence from ED symptomatology). Hence, additional longitudinal studies collecting follow-up data are needed to determine the long-term effect of the associations found.

Notwithstanding these limitations, the study also has several strengths that should be noted. To the best of our knowledge, this is the first study examining the potential role of BMI changes across the lifespan in the phenotypic characteristics and severity of patients with ED, as well as their association with therapeutic response.

## 5. Conclusions

In short, our findings provide support for considering the course of BMI as a transdiagnostic feature that serves as a possible marker of severity and treatment outcome. Our findings corroborate that lifetime obesity is associated with greater general psychopathology and with some personality traits such as low persistence and self-directedness, and high reward dependence (i.e., low self-confidence). Thus, more functional scores on these personality traits may act as protective factors against weight gain. Finally, a relevant finding from our research reveals that only a subgroup of patients with BED (namely, those in the OB-OB BMI profile) have significantly less treatment adherence and higher dropout rates, which might be because they do not link their obesity to the ED. Thus, the findings derived from this study might improve our ability to identify clinical features related to the symptomatic expression and prognosis of these patients (namely weight changes) and, thereby, aid in tailoring the best treatment targets.

## Figures and Tables

**Figure 1 nutrients-13-02034-f001:**
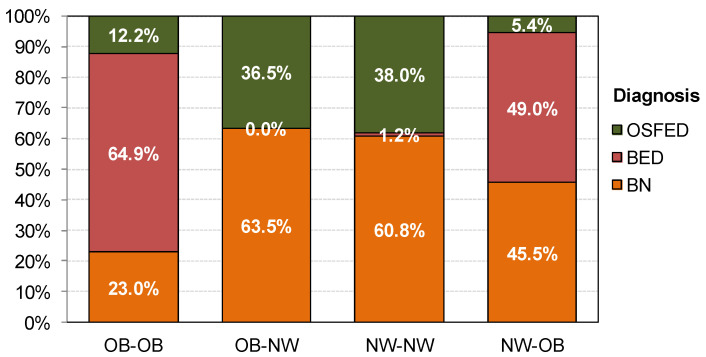
Distribution of the diagnostic subtype within the lifetime BMI profiles. Note: OB: obesity; NW: normal weight; BN: bulimia nervosa; BED: binge eating disorder; OSFED: other specified feeding or eating disorder.

**Figure 2 nutrients-13-02034-f002:**
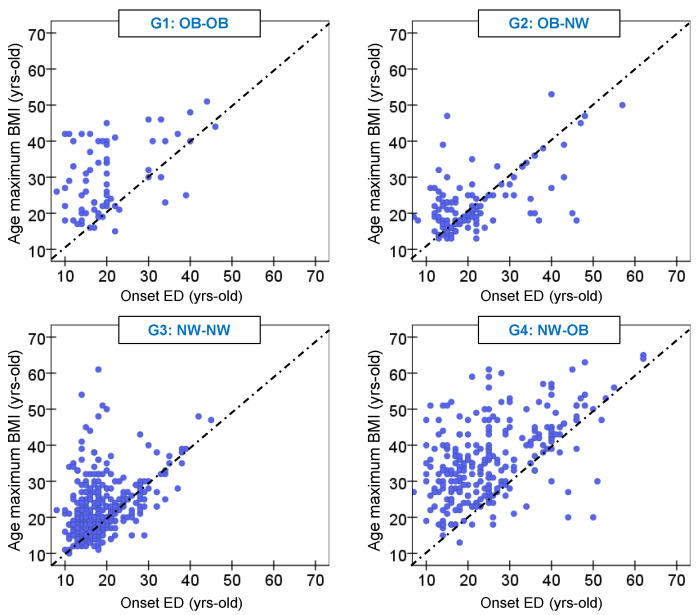
Scatterplot with the age of onset of the ED and the age of the maximum BMI.

**Figure 3 nutrients-13-02034-f003:**
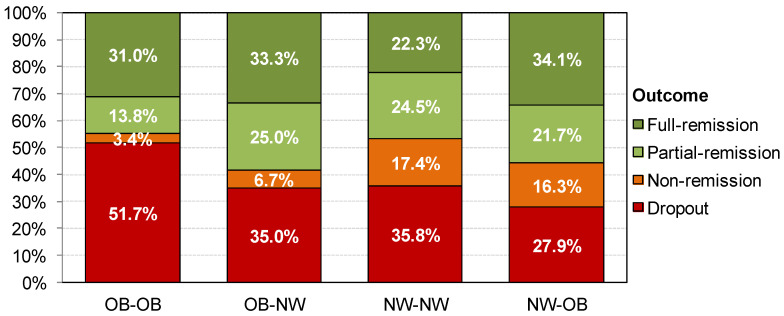
Distribution of the CBT outcome within the lifetime BMI profiles. Note: OB: obesity; NW: normal weight.

**Table 1 nutrients-13-02034-t001:** Comparison between the BMI profiles on sociodemographics, clinical, and motivational measures.

	OB-OB *n* = 74		OB-NW *n* = 156		NW-NW *n* = 756		NW-OB *n* = 314		OB-OB vs OB-NW		OB-OB vs NW-NW		OB-OB vs NW-OB		OB-NW vs NW-NW		OB-NW vs NW-OB		NW-NW vs NW-OB	
	*n*	%	*n*	%	*n*	%	*n*	%	*p*	*|h|*	*p*	*|h|*	*p*	*|h|*	*p*	*|h|*	*p*	*|h|*	*p*	*|h|*
Diagnosis BN	17	23.0%	99	63.5%	460	60.8%	143	45.5%	**<0.001**	**0.90** ^†^	**<0.001**	**0.83 ^†^**	**0.001**	**0.52 ^†^**	0.355	0.05	**<0.001**	0.37	**<0.001**	0.31
BED	48	64.9%	0	0.0%	9	1.2%	154	49.0%		**1.92** **^†^**		**1.84 ^†^**		0.32		0.16		**1.39** **^†^**		**1.32 ^†^**
OSFED	9	12.2%	57	36.5%	287	38.0%	17	5.4%		**0.59** **^†^**		**0.62 ^†^**		0.24		0.03		**0.83** **^†^**		**0.86** **^†^**
Gender Female	64	86.5%	124	79.5%	734	97.1%	296	94.3%	0.199	0.19	**<0.001**	0.39	**0.020**	0.27	**<0.001**	**0.57** **^†^**	**<0.001**	0.45	**0.027**	0.14
Male	10	13.5%	32	20.5%	22	2.9%	18	5.7%												
Education Primary	42	56.8%	89	57.1%	294	38.9%	156	49.7%	0.969	0.01	**0.009**	0.36	0.209	0.14	**<0.001**	0.37	**0.041**	0.15	**0.002**	0.22
Secondary	26	35.1%	53	34.0%	344	45.5%	108	34.4%		0.02		0.21		0.02		0.24		0.01		0.23
University	6	8.1%	14	9.0%	118	15.6%	50	15.9%		0.03		0.23		0.24		0.20		0.21		0.01
Civil status Single	42	56.8%	111	71.2%	654	86.5%	125	39.8%	**0.003 ***	0.30	**<0.001**	**0.70** **^†^**	**0.026**	0.34	**<0.001**	0.38	**<0.001**	**0.66** **^†^**	**<0.001**	**1.11** **^†^**
Partner	27	36.5%	26	16.7%	68	9.0%	151	48.1%		**0.51** **^†^**		**0.69 ^†^**		0.24		0.23		**0.71 ^†^**		**0.96 ^†^**
separated	5	6.8%	19	12.2%	34	4.5%	38	12.1%		0.19		0.10		0.18		0.28		0.00		0.28
Unemployed	25	33.8%	63	40.4%	202	26.7%	160	51.0%	0.341	0.14	**<0.001**	0.15	**<0.001**	0.35	**<0.001**	0.29	**<0.001**	0.21	**<0.001**	**0.51 ^†^**
Student	18	24.3%	43	27.6%	385	50.9%	24	7.6%		0.07		**0.57 ^†^**		**0.51 ^†^**		**0.52 ^†^**		**0.54 ^†^**		**1.08 ^†^**
Employed	31	41.9%	50	32.1%	169	22.4%	130	41.4%		0.20		0.43		0.01		0.22		0.19		0.42
BMI measures	Mean	SD	Mean	SD	Mean	SD	Mean	SD	*p*	*|d|*	*p*	*|d|*	*p*	*|d|*	*p*	*|d|*	*p*	*|d|*	*p*	*|d|*
BMI current	44.95	9.46	22.16	1.74	20.81	1.66	36.54	5.25	**<0.001**	**3.35 ^†^**	**<0.001**	**3.55 ^†^**	**<0.001**	**1.10 ^†^**	**<0.001**	**0.80 ^†^**	**<0.001**	**3.67 ^†^**	**<0.001**	**4.04 ^†^**
BMI max.	48.45	9.50	34.72	4.94	22.76	1.43	38.48	5.79	**<0.001**	**1.81 ^†^**	**<0.001**	**3.78 ^†^**	**<0.001**	**1.27 ^†^**	**<0.001**	**3.29 ^†^**	**<0.001**	**0.70 ^†^**	**<0.001**	**3.73 ^†^**
BMI min.	33.41	3.17	19.92	2.55	17.99	1.88	21.89	2.17	**<0.001**	**4.69 ^†^**	**<0.001**	**5.92 ^†^**	**<0.001**	**4.24 ^†^**	**<0.001**	**0.86 ^†^**	**<0.001**	**0.83 ^†^**	**<0.001**	**1.92 ^†^**
Motivation: Intensity	6.14	1.62	5.64	2.17	5.12	1.97	6.24	1.75	0.069	0.26	**<0.001**	**0.56 ^†^**	0.667	0.06	**0.002**	0.25	**0.001**	0.31	**<0.001**	**0.60 ^†^**
Need treatment	6.74	1.48	5.87	2.28	5.71	2.18	6.74	1.66	**0.003**	**0.52 ^†^**	**<0.001**	**0.56 ^†^**	0.987	0.00	0.361	0.07	**<0.001**	0.44	**<0.001**	**0.53 ^†^**
Social impairment	5.18	2.56	4.90	2.55	4.69	2.36	5.32	2.36	0.410	0.11	0.099	0.20	0.630	0.06	0.335	0.08	0.069	0.17	**<0.001**	0.27
Self-concern	7.00	1.09	6.15	2.18	6.13	2.14	6.96	1.59	**0.003**	**0.51 ^†^**	**<0.001**	**0.51 ^†^**	0.881	0.03	0.871	0.01	**<0.001**	0.42	**<0.001**	0.44
Family concern	6.09	2.32	6.57	2.09	6.68	2.00	5.95	2.36	0.112	0.22	**0.023**	0.27	0.595	0.06	0.553	0.05	**0.003**	0.28	**<0.001**	0.33

Note: OB: obesity; NW: normal weight; BN: bulimia nervosa; BED: binge eating disorder; OSFED: other specified feeding and eating disorder; SD: standard deviation; * bold: significant comparison; † bold: effect size into the range mild/moderate (|*d*| > 0.50 or |*h*| > 0.50) to large/high (|*d*| > 0.80 or |*h*| > 0.80).

**Table 2 nutrients-13-02034-t002:** Comparison between the groups for the clinical profile.

	OB-OB *n* = 74		OB-NW *n* = 156		NW-NW *n* = 756		NW-OB *n* = 314		OB-OB vs OB-NW		OB-OB vs NW-NW		OB-OB vs NW-OB		OB-NW vs NW-NW		OB-NW vs NW-OB		NW-NW vs NW-OB	
	Mean	SD	Mean	SD	Mean	SD	Mean	SD	*p*	*|d|*	*p*	*|d|*	*p*	*|d|*	*p*	*|d|*	*p*	*|d|*	*p*	*|d|*
Age (years-old)	32.70	10.56	29.12	9.01	24.91	7.60	37.61	10.77	**0.004 ***	0.36	**0.001 ***	**0.85 ^†^**	**0.001 ***	**0.51 ^†^**	**0.001 ***	**0.51 ^†^**	**0.001 ***	**0.86 ^†^**	**0.001 ***	**1.36 ^†^**
Onset of ED	19.91	8.84	20.88	8.74	17.76	4.88	25.28	10.82	0.356	0.11	**0.018**	0.30	**0.001 ***	**0.54 ^†^**	**0.001 ***	0.44	**0.001 ***	0.45	**0.001 ***	**0.90 ^†^**
Duration of ED	12.80	8.62	8.27	7.21	7.19	6.95	12.41	10.01	**0.001 ***	**0.57 ^†^**	**0.001 ***	**0.72 ^†^**	0.708	0.04	0.122	0.15	**0.001 ***	**0.51 ^†^**	**0.001 ***	**0.61 ^†^**
Binges/week	3.98	4.94	3.97	6.13	4.11	5.77	6.16	5.59	0.988	0.00	0.850	0.02	**0.003 ***	0.41	0.775	0.02	**0.001 ***	0.37	**0.001 ***	0.36
Vomits/week	1.63	4.67	7.04	9.40	5.16	7.38	2.46	6.14	**0.001 ***	**0.73 ^†^**	**0.001 ***	**0.57 ^†^**	0.376	0.15	**0.003 ***	0.22	**0.001 ***	**0.58 ^†^**	**0.001 ***	0.40
EDI-2: Total score	113.26	38.67	109.56	45.35	100.72	40.28	111.45	38.26	0.517	0.09	**0.011 ***	0.32	0.729	0.05	**0.013**	0.21	0.633	0.05	**0.001 ***	0.27
SCL-90R GSI	1.73	0.73	1.87	0.75	1.72	0.70	1.91	0.73	0.161	0.19	0.875	0.02	0.059	0.24	**0.014 ***	0.21	0.638	0.04	**0.001 ***	0.26
SCL-90R PST	62.12	16.26	65.42	18.65	64.78	17.00	66.32	15.82	0.167	0.19	0.196	0.16	0.054	0.26	0.669	0.04	0.583	0.05	0.174	0.09
SCL-90R PSDI	2.44	0.55	2.49	0.60	2.29	0.53	2.52	0.56	0.484	0.09	**0.032 ***	0.27	0.252	0.15	**0.001 ***	0.35	0.615	0.05	**0.001 ***	0.41
Novelty seeking	101.88	15.70	99.28	17.08	104.22	15.38	101.97	15.76	0.242	0.16	0.222	0.15	0.965	0.01	**0.001 ***	0.30	0.081	0.16	**0.033 ***	0.14
Harm avoidance	121.05	18.82	116.81	20.07	114.81	19.27	122.98	18.02	0.115	0.22	**0.007 ***	0.33	0.434	0.10	0.232	0.10	**0.001 ***	0.32	**0.001 ***	0.44
Reward dependence	104.08	17.36	99.02	14.53	102.35	14.85	102.31	15.68	**0.018 ***	0.32	0.348	0.11	0.366	0.11	**0.013 ***	0.23	**0.027 ***	0.22	0.969	0.00
Persistence	103.64	17.22	106.80	21.26	110.63	19.65	104.18	20.87	0.263	0.16	**0.004 ***	0.38	0.833	0.03	**0.030 ***	0.19	0.182	0.12	**0.001 ***	0.32
Self-directedness	114.07	19.96	113.45	19.45	115.94	19.61	111.05	20.23	0.824	0.03	0.436	0.09	0.238	0.15	0.151	0.13	0.216	0.12	**0.001 ***	0.25
Cooperativeness	132.88	18.23	130.37	16.32	133.36	15.23	132.57	16.65	0.264	0.14	0.803	0.03	0.881	0.02	**0.033 ***	0.19	0.158	0.13	0.458	0.05
Self-transcendence	68.95	15.31	64.28	14.21	64.98	14.20	66.58	15.82	**0.024 ***	0.32	**0.027 ***	0.27	0.213	0.15	0.585	0.05	0.109	0.15	0.104	0.11

Note: OB: obesity; NW: normal weight; SD: standard deviation; * bold: significant comparison; ^†^ bold: effect size into the range mild/moderate (|*d*| > 0.50) to large/high (|*d*| > 0.80).

**Table 3 nutrients-13-02034-t003:** Comparison between the groups for impulsivity measures.

	OB-OB G1; *n* = 74		OB-NW G2; *n* = 156		NW-NW G3; *n* = 756		NW-OB G4; *n* = 314		G1 vs G2		G1 vs G3		G1 vs G4		G2 vs G3		G2 vs G4		G3 vs G4	
	n	%	n	%	n	%	n	%	*p*	*|h|*	*p*	*|h|*	*p*	*|h|*	*p*	*|h|*	*p*	*|h|*	*p*	*|h|*
NSSI	18	24.3%	60	38.5%	257	34.0%	90	28.7%	**0.034 ***	0.31	0.092	0.21	0.454	0.10	0.286	0.09	**0.032 ***	0.21	0.090	0.12
Suicidal ideation	37	50.0%	85	54.5%	367	48.5%	181	57.6%	0.524	0.09	0.811	0.03	0.233	0.15	0.177	0.12	0.516	0.06	**0.007 ***	0.18
Suicidal attempts	16	21.6%	39	25.0%	147	19.4%	80	25.5%	0.575	0.08	0.653	0.05	0.489	0.09	0.117	0.13	0.911	0.01	**0.028 ***	0.14
Alcohol abuse	1	1.4%	16	10.3%	77	10.2%	24	7.6%	**0.016 ***	0.39	**0.013 ***	0.39	**0.047 ***	0.31	0.979	0.00	0.339	0.09	0.195	0.09
Drugs abuse	8	10.8%	28	17.9%	157	20.8%	41	13.1%	0.164	0.20	**0.041 ***	0.28	0.601	0.07	0.425	0.07	0.158	0.14	**0.003 ***	0.21

Note: OB: obesity; NW: normal weight; NSSI: nonsuicidal self-injury; vs: versus; * bold: significant comparison.

**Table 4 nutrients-13-02034-t004:** Age of onset of ED and age of maximum BMI.

	OB-OB G1; *n* = 69		OB-NW G2; *n* = 125		NW-NW G3; *n* = 558		NW-OB G4; *n* = 260		G1 vs G2		G1 vs G3		G1 vs G4		G2 vs G3		G2 vs G4		G3 vs G4	
Maximum BMI	*n*	%	*n*	%	*n*	%	*n*	%	*p*	*|h|*	*p*	*|h|*	*p*	*|h|*	*p*	*|h|*	*p*	*|h|*	*p*	*|h|*
Previous onset ED	7	10.1%	44	35.2%	137	24.6%	21	8.1%	**0.001 ***	**0.63 ^†^**	**0.001 ***	0.39	0.564	0.07	**0.008 ***	0.23	**0.001 ***	**0.70 ^†^**	**0.001 ***	**0.51 ^†^**
Coincides onset ED	5	7.2%	34	27.2%	128	22.9%	12	4.6%		**0.55 ^†^**		**0.51 ^†^**		0.11		0.10		**0.65 ^†^**		**0.55 ^†^**
Posterior onset ED	57	82.6%	47	37.6%	293	52.5%	227	87.3%		**1.03 ^†^**		**0.68 ^†^**		0.13		0.30		**1.20 ^†^**		**0.82 ^†^**

Note: OB: obesity; NW: normal weight; vs: versus; * bold: significant comparison; ^†^ bold: effect size into the range mild/moderate (|*h*| > 0.50) to large/high (|*h*| > 0.80).

**Table 5 nutrients-13-02034-t005:** Comparison for the CBT outcomes.

	OB-OB G1; *n* = 29		OB-NW G2; *n* = 60		NW-NW G3; *n* = 282		NW-OB G4; *n* = 129		G1 vs G2		G1 vs G3		G1 vs G4		G2 vs G3		G2 vsG4		G3 vs G4	
	*n*	%	*n*	%	*n*	%	*n*	%	*p*	*|h|*	*p*	*|h|*	*p*	*|h|*	*p*	*|h|*	*p*	*|h|*	*p*	*|h|*
Drop out	15	51.7%	21	35.0%	101	35.8%	36	27.9%	0.408	0.34	**0.038 ***	0.32	**0.040 ***	**0.51 ^†^**	0.105	0.02	0.290	0.15	0.081	0.17
Non-remission	1	3.4%	4	6.7%	49	17.4%	21	16.3%		0.15		**0.50 ^†^**		0.44		0.33		0.31		0.03
Partial-remission	4	13.8%	15	25.0%	69	24.5%	28	21.7%		0.29		0.27		0.21		0.01		0.08		0.07
Full-remission	9	31.0%	20	33.3%	63	22.3%	44	34.1%		0.05		0.20		0.07		0.25		0.02		0.26

Note: OB: obesity; NW: normal weight; vs: versus; * bold: significant comparison; ^†^ bold: effect size into the range mild/moderate (|*h*| > 0.50) to large/high (|*h*| > 0.80).

## Data Availability

Data are not available in any repository. Contact with corresponding authors.

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
