# Peer review of "Lifetime Weight Course as a Phenotypic Marker of Severity and Therapeutic Response in Patients with Eating Disorders"

_nutrients, 2021, doi:10.3390/nu13062034_

Round 1

Reviewer 1 Report

This study compared ED severity and treatment outcomes among four specific clusters based on BMI-changes across the lifespan. They found that lifetime obesity is associated with greater general psychopathology and with some personality traits such as low persistence and self-directedness, and high reward dependence. Also, greater extreme weight changes (NW-OB and OB-NW) were associated with higher psychopathology, but not with greater ED severity. An issue was that higher dropout rates were found in the OB-OB group. 

This is an excellent study and well presented as it is. There are some issues that needs to be addressed:

  • patients reported weight changes starting before adulthood, while inclusion criteria was being adult (at or above 18 years). The reliability of this needs to be addressed e.g. by comparing weight changes reported before adulthood  with reported weight changes in adulthood only. This could e.g. be split into two separate analyses.
  • current overweight defined as BMI25-29.9 may have influenced the results. This may imply a selection of individuals with more psychopathology, according to the current study results. It would be prudent to include a more extensive clinical and scientific justification for excluding this group, and, preferably also a group comparison to defend exclusion this group of individuals.
  • The Finner method for adjusting for multiple comparisons is to me completely novel and after searching the internet, not mentioned as a specific method called the "Finner method". The original publication can be found however, it is not clear that this is a proposed method. The authors should include a more elaborate explanation to the choice of this "method" and the rationale for not using other methods. It would also be interesting to see how the results would change if using the classic Bonferroni method. Clearly, there is a range of statistical analyses done which increases the likelihood of type 1 error. 
  • the number of dropouts is an issue in the OB-OB group. This may clearly have impacted the results. I would propose the authors include an additional analysis of the psychopathology of the different groups in relation to weigh changes over time adjusting for drop-outs.
  • Has the motivational instrument been validated for use in these populations (obesity) and albeit being a VAS scale, has it been used/validated in this language (Spanish)? And are there population based normalized values for this instrument? Also, has it been used to assess changes before and after intervention? 

Reviewer 2 Report

This manuscript investigated cross-sectional clinical and personality correlates and treatment outcomes of four distinct BMI-change clusters in treatment-seeking persons with an eating disorder who presented with a minimally healthy body weight. A strength of this study was a large sample of treatment-seeking adults with eating disorders. On the other hand, my enthusiasm is limited by the lack of focus in the manuscript. For instance, a great deal of information is presented in the manuscript, and I was left wondering what is the most important to take away. Perhaps this could be information about treatment outcomes as a function of weight history?  

  1. The authors provide a detailed literature review on weight history and weight suppression in the introduction, but do not provide hypotheses. Based on what is known, what did the authors hypothesize they might find?
  2. What was the sample size prior to exclusion? A CONSORT-type diagram showing admissions and why patients were excluded from the study would be helpful.
  3. I think it is important to state in the text how the participants who were and were not included in the treatment outcomes aim of this paper differed. For instance, they differed by diagnosis, sex, education level, and employment status. These do not seem like trivial factors to me.
  4. Why did the authors exclude persons with a body mass index below the minimally healthy threshold? Why did the authors exclude persons with overweight? More detail is needed about these decisions.
  5. Similarly, why did the authors not exclude persons with a history of underweight?
  6. Who administered the SCID to patients? For instance, trained assessors, psychiatrists, psychologists, etc.? Was there any kind of consensus for the SCID diagnoses?
  7. How were “impulsive behaviors” assessed?
  8. In the results, the authors present findings in both tables and text, which is redundant.
  9. Overall, a lot of information is presented and it is hard for the reader to pinpoint the take-home messages. I think the authors could consider focusing only on treatment outcomes for the purpose of this paper.

Round 2

Reviewer 1 Report

I find that the authors have addressed almost all of the comments I have had on this manuscript. The only issue they have not addressed is on the number of dropouts in the OB-OB group and the requested analysis on taking drop-outs into consideration when comparing the groups. It is clear that data are missing on the drop-outs but that does not preclude adjusting for number of drop-outs in group comparisons. I suggest the authors either defend why not addressing this or adding an analysis adjusting for drop-outs.

Author Response

Apologies because we are not quite sure we understood what the reviewer's request. To address the suggestion of the reviewer, we have now re-run the comparison between the groups with the following considerations:

  1. The cross-sectional analyses (displayed in Tables 1-2-3-4 of the manuscript) were additionally performed for the participants with data available for the treatment (N=500). These results are now included in the new Table S2 (please see the supplementary material attached). The analyses provide similar results to those obtained for the whole sample at baseline (N=1,300).
  2. The cross-sectional analyses (displayed in Tables 1-2-3-4 of the manuscript) were also performed for the completers subsample (participants with data available in the treatment and did not dropout, N=327). These results are included in the new Table S3 (supplementary material), with values also similar to those obtained for the whole sample at baseline (N=1,300).
  3. The comparison between the groups for the treatment outcome (displayed in Table 5) was also performed using a weight procedure, defining sampling weights inversely proportional to the probability of participation in the longitudinal substudy (the estimation of the concrete weight-factor was based on the flow-chart displayed in the Figure S1). The results of this additional comparison are displayed in new Table S4 (supplementary material), and are also similar to those obtained without weighting cases.

However, if the reviewer requested something different, please let us know, as we are willing to make any changes to improve the manuscript.

Reviewer 2 Report

I thank the authors for their responsiveness to my suggestions. I think the authors misunderstood my question of why the authors opted not to exclude persons with a history of underweight (rather than current underweight). That is, why did the authors include persons who may have been underweight (BMI < 18.5 kg/m2) in the past? Otherwise, I have no further comments.

Author Response

We apologize for the misinterpretation of the question. We excluded patients who reported a lifetime diagnosis of anorexia nervosa (AN). But admittedly, we did not exclude participants who may have been underweight (BMI <18.5 kg/m2) in the past because of the difficulty of interpreting retrospective reports of age-associated to BMI changes and the unavailability of previous height data. However, we understand the reviewer's concern and agree with what he/she points out. Therefore, we have included the following statement in the limitation section: “Participants who may have been underweight (BMI < 18.5 kg/m2) in the past were did not excluded in this study because of the difficulty of interpreting retrospective reports of age-associated BMI changes and the unavailability of height data. Further studies should exclude participants with a lifetime BMI less than 18.5 by controlling for weight and height at each time point.”